# Parents’ and Children’s Experiences with a Coordinating Professional in Integrated Care for Childhood Overweight and Obesity—A Novel Dutch Approach

**DOI:** 10.3390/ijerph19105797

**Published:** 2022-05-10

**Authors:** Sanne A. A. De Laat, Monique A. M. Jacobs, Edgar G. Van Mil, Ien A. M. Van de Goor

**Affiliations:** 1Tranzo, Scientific Center for Care and Wellbeing, Tilburg School of Social and Behavioral Sciences, Tilburg University, P.O. Box 90153, 5000 LE Tilburg, The Netherlands; s.de.laat@ggdhvb.nl; 2Municipal Health Service (GGD) Hart voor Brabant, P.O. Box 3024, 5003 DA Tilburg, The Netherlands; mo.jacobs@ggdhvb.nl; 3Jeroen Bosch Hospital, Henri Dunantstraat 1, 5223 GZ ‘s-Hertogenbosch, The Netherlands; e.g.vanmil@jbz.nl; 4Brightlands Campus, Maastricht University, Greenport, Villafloraweg 1, 5928 SZ Venlo, The Netherlands

**Keywords:** coordinating professional, prevention, integrated care, qualitative interviews, online questionnaire, experiences, parents, children’s overweight, obesity

## Abstract

Background: In the new integrated program of care for childhood overweight and obesity (ICCO), a Youth Health Care (YHC) nurse has the role of a coordinating professional. After a broad assessment of strengths and weaknesses in the family setting, this coordinating professional makes a plan of action with the child and parents and involves other professionals when needed. The aim of this study was to explore the experiences of parents and children with the coordinating professional in the ICCO. Material & Methods: Semi-structured interviews were conducted with eight families. Interview data were analyzed using content analysis. In addition, descriptive data on involved professionals and referrals was collected with an online questionnaire in 38 families. Results: In total, eight families (8 mothers, 2 fathers, four boys and three girls aged 10–12 yrs) were interviewed and 38 children and parents filled in (three consecutive) online questionnaires. Findings: The main themes related to the experiences of parents and children with the CP: parents and children felt supported and understood by the coordinating professional. They appreciated the broad perspective and personal approach. Contacts with the coordinating professional were not always frequent. Major points of improvement concerned the intensity of the follow-up and collaboration. Only few parents experienced collaboration between the coordinating professional and other professionals in the ICCO. Conclusions: Parents and children appreciated the personal approach of the Youth Health Care nurse as a CP. The role of the coordinating professional, however, appears not fully implemented yet. Strengthening the promising role of the coordinating professional in the ICCO is recommended.

## 1. Introduction

Childhood obesity and overweight are a worldwide problem with a large impact on health and well-being [1]. In the Netherlands about 1 out of 7 of children aged 2–19 years has either overweight or obesity [2]. Obesity during childhood is an important risk factor for the development of various comorbidities including dyslipidemia, hypertension and diabetes. The presence of comorbidities is already evident in primary school children with obesity [3]. In addition to the effects on biomedical health, overweight and obesity may affect the quality of life of young people causing psychological problems [4]. Early recognition and interventions are recommended. Unfortunately, interventions are often only slightly effective [5]. Most families with children with obesity know what a healthy lifestyle for their children should be and are willing to invest in the future health of their children but are often not able to provide this lifestyle. The challenge for the health care professional is to identify the underlying problems for each child and family. This knowledge can help the family solve the problems or work around issues. Together with the family, the professional can make a plan of action that fits the needs of the family. This may lead to an increase in the parents’ intrinsic motivation to take on the challenge of lifestyle changes [6].

A personalized and integrated approach is wanted, especially for children with overweight and obesity and parents in complex and vulnerable circumstances. Children from low socioeconomic status (SES) families are more likely to develop overweight and obesity. In addition to low SES, other risk factors for childhood obesity are: a lack of social support, parental unemployment, belonging to a minority group, having a migrant background and adverse childhood experiences [7]. Families who are coping with complex circumstances are especially likely to encounter more barriers than facilitators to their participation in and completion of lifestyle interventions [8].

In the Netherlands, a national model for integrated care for childhood overweight and obesity (ICCO) was developed in cooperation with eight municipalities, combining experiences from science, policy and practice [9,10,11]. Based on the ICCO model, more municipalities implemented promising local integrated approaches. In addition, several studies have started to evaluate the effectiveness and implementation process of the ICCO [12,13,14].

In the ICCO, families receive support from a coordinating professional (CP), a role performed by trained YHC nurses. The intention is to gradually empower the family and decrease the help they need from professionals. After approximately one year of guidance by a CP, the family members should be able to maintain a healthier lifestyle by themselves [10].

The CP collaborates with professionals from the medical, social and public sectors in order to provide the necessary stepped and matched help for the families. The CP functions as a connector between the sectors. In addition, a CP functions as a coach for the family, she tries to improve the knowledge of the parents and supports parents to develop and sustain a healthy lifestyle fitting the families’—sometimes complex—situation [11].

According to our previous study, in the city of ‘s-Hertogenbosch, YHC nurses believe that the role of CP fits their profession. However, some improvements in the implementation of the role of the coordinating professional are still desired [15,16].

The aim of this study is to explore the experiences of parents and children with the role of the YHC nurse as CP in the ICCO, to explore how the new approach is appreciated and to describe points for improvement. It is important to know how the new approach is received by the families. This is the first study exploring the experiences of parents and children with the CP in the ICCO in the Netherlands.

## 2. Materials & Methods

### 2.1. Research Team and Reflexivity

The principal researcher, female, MD, youth health care physician and PhD researcher and trained in both qualitative and quantitative research methods, carried out the interviews in collaboration with research assistants. The researcher introduced herself as a PhD researcher and youth health care physician.

### 2.2. Study Design

This study is part of a larger study in which the implementation and effectiveness of the ICCO in ‘s-Hertogenbosch, a medium sized city in the Netherlands, is evaluated [12]. The larger study is a mixed methods study: a qualitative study with semi-structured interviews among parents and children about their experiences with the implementation of this innovative approach of the CP, and an effectiveness study consisting of online questionnaires at three consecutive moments in time [17] where the families receiving ICCO were compared to a control group of families receiving care as usual.

### 2.3. Data Collection

In this article focusing on families’ experiences with the CP in the ICCO, both qualitative and quantitative data were used. Participants were selected from the respondents that had agreed to be contacted for an interview in the questionnaire–study. They were approached by telephone for face-to-face interviews. In total 8 families, represented by 8 mothers, 2 fathers, 4 girls and 3 boys, participated in the interviews. Interviews were held in the home setting. There were sufficient participants from a non-Western background. Informed consent was obtained from all participants (parents and children ≥ 12 years old).

An interview guide with open questions and statements about the role of the YHC nurse as CP and the ICCO was used. Interviews were conducted in Dutch by a researcher and a research assistant, starting together with the parent(s) and the child. After the first part of the interview, the parent(s) and child were interviewed separately. Interviews were ended together by formulating a letter to the CP with tips and tops. Demographic variables like age, profession of the parents and family size were noted. At the end of the interview participating families received a gift voucher of 10 euros.

The third questionnaire of the effectiveness study [12] included questions on experienced support, on which professionals were involved last year and on referrals by YHC and the general practitioner. Parents were asked how they appreciated the care by the YHC last year; they were asked to give a grade between 0–10.

### 2.4. Data Analysis Strategies

The interviews were guided by a topic list (see Appendix A). Interviews were recorded and transcribed verbatim. Transcriptions were coded thematically and openly. Themes from the topic list were adapted inductively by the data through content analysis. In the first step, the researcher and research assistant familiarized themselves with the data by checking the transcripts. In the second step, a selection of initial codes was formulated based on the first interviews and formulated research questions. The coding was performed using ATLAS.ti version 8 for Windows. (ATLAS.ti Scientific Software Development GmbH, Berlin, Germany). The researcher and research assistant compared their codes to ensure consistency during the coding process. During the coding process, extra codes were added if needed. In the third step overarching themes were formulated based on clustering of the codes. These themes were discussed in the research group [18].

Participant quotes are presented to illustrate findings and themes. From the online questionnaire only descriptive data were used.

### 2.5. Ethical Issues

The study was approved by the Medical Ethics Committee Brabant (reference number METC P1737). The approval of the Medical Ethics Committee covered all other aspects in the context of medical scientific research in children.

## 3. Results

The online questionnaire was completed by 38 parents in the ICCO group and by 20 parents in a control group receiving care as usual. Questionnaires were returned between October 2018 and January 2020. After participating in the effectiveness study, 22 parents in the ICCO group in ‘s-Hertogenbosch received an e-mail with an invitation for an interview about their experiences in the ICCO. Six other families involved in ICCO were asked to participate in an interview during a visit at the Jeroen Bosch Hospital. Interested parents were called by a research assistant. In total eight families agreed to participate. An appointment was made for an interview at home (*n* = 6) or at the public health office (*n* = 1). The last interview was held by video phone call because of COVID-19 limitations (*n* = 1). Interviews were held between November 2019 and March 2020. The duration of the interviews varied between 49 and 84 min. The average duration was 60 min.

Demographic characteristics of participants in the ICCO are presented in Table 1. The interview group consisted of six families from the ICCO questionnaire group and two families recruited by the hospital. Together, eight mothers, two fathers, four girls and three boys (aged 10–12 years old) participated in the interviews.

On average, children in the interviewed families were older and there were more children of whom both parents were not born in the Netherlands, compared with participants who completed the questionnaire.

### 3.1. Findings

Main themes emerging from the data analysis related to experiences of parents and children with the CP. The first main theme was the overall positive appreciation of the CP such as feeling supported and understood, feeling familiar with the YHC nurse in this role, with a minor point being that contact was not always frequent or followed up. A second main theme were the children’s experiences which were also positive such as feeling motivated and advised by the CP, but they were less outspoken. A third main theme focused on the collaboration in the professional network (ICCO). Parents did not experience much (or intensive) collaboration between involved health care professionals. They did experience and appreciate the increased attention for creating healthy environments at their children’s schools.

### 3.2. Experiences of Parents with the CP

#### 3.2.1. Evaluating the Role of the CP

During the interview, statements about the role of the CP were discussed with the parents. All parents felt supported by the CP. They appreciated that the CP was aware of the background and history of the families because of her regular job at the YHC center as a YHC nurse. There was a broad assessment at the start of the support in almost all families. All parents felt that they could talk about any topic with the CP, also about topics that seemed not directly related to overweight. All parents felt that the nurses had enough time for them. Parents mentioned that the CP had a broad perspective on the overweight problem and did not only give advice about food and exercise.

“*Yes, I’ve definitely noticed that. Based on her questions and interest, it’s not just about what she eats all day, but how does she feel. Yes, so I have noticed that*.” (Mother 2)

Parents were asked which professionals they considered to be in the best position to take the role of CP. Most parents think that this role fits the YHC nurse best. Mainly because the YHC nurse knows the child’s history, is familiar, easily accessible and easy to talk to.

Most of the families did not have very intensive or frequent contact with their CP last year. However, they knew how to contact the CP when they had questions.


*Interviewer: “And have you had any contact with [name YHC nurse] recently, the YHC nurse?” Father: “Hmm no.” Mother: “No, because he said that if there is anything, I can always call him. So yes, you can. I ran into him yesterday”. Interviewer: “In the neighborhood here?” Mother: “Yes, yes.” *
(Mother and father 3)

In some of the interviewed families the last contact with the CP was already a long time ago, without any planned follow-up. One of the mothers recounted that there was no follow-up arranged after the CP left and started a new job elsewhere. Another mother was grateful that the CP arranged financial support so that her daughter was able to play tennis but did not know where to go now.

“*Yes, because it was a problem for me, because at a certain point, [YHC nurse] had left and the Youth Sport Fund had to be arranged. Then I thought, oh my God, where do I have to go now? However, fortunately, there is also someone at school who can arrange that.*”(Mother 6)

#### 3.2.2. Appreciation of the YHC Nurse as a CP

Interviewed parents were asked to give the YHC nurse a grade between 0–10. High grades were given, with a mean of 8.8. The six interviewed parents who also completed the questionnaire gave higher grades during the interviews than in the questionnaire (8.8 versus 7.5). In the questionnaire the other parents in the ICCO group (those who were not interviewed) gave a mean of 6.2.

*Interviewer: “If you had to give the YHC nurse a grade, what grade would you give him?*”

*Mother: “Yes, for me it would be a 10.*”

*Interviewer: “Yes, tell me, why would you give him a 10?*”

*Mother: “Yes, because they are doing really well. They don’t give you the feeling like hey, I know everything, and you do everything wrong. Yes.*” (Mother 3)

### 3.3. Experiences of Children with the CP

Most interviewed children appreciated the contact with the YHC nurse as a CP. They felt understood and supported. They felt better after contact with the CP. Some children felt motivated and remembered being given advice about healthy lifestyles and sports.

“*Yes, he listens well, he also understands me with many things. I liked it.*” (Child 4)

“*The conversations we had were really nice and I felt understood really well.*”(Child 1)

### 3.4. Experiences of Families with the ICCO

#### 3.4.1. Involved Professionals

Interviewed families were supported by six different CPs and some also by other professionals. According to the parents, the general practitioner is not always well informed by the CP about the support initiated for the family in the ICCO. Interviewed families had more contact with a pediatrician, social worker and physiotherapist (with an exercise program for children with overweight) as compared with the families who completed the questionnaire. Most parents in the ICCO who completed the questionnaire did not experience intensive contact with the YHC nurse as a CP. The dietician was involved in 32% of the children, the pediatrician in 21%. See Table 2.

#### 3.4.2. Collaboration between Professionals

An important component of the ICCO is building a strong network in which professionals collaborate in order to provide matched and stepped care for families. The CP is supposed to be a connector between different professionals in the network. Three interviewed families did experience collaboration between professionals involved in their family and were content with this collaboration.

“*At the hospital they check and advise and if I want help, they work together with [child welfare organisation] as well. [Child welfare organisation] and the school and psychiatrist for [name interviewed child] and [name other daughter] have been to the psychiatrist in [municipality]. She [CP] has contacted all these people.*”(Mother 7)

Most families (5 out of 8) had not explicitly experienced collaboration between the CP and other professionals. Three families expected that the CP would collaborate with other professionals if necessary. In two families the CP was the only supporting professional for the family, no other professionals were involved. One of the mothers would have appreciated to receive more information about the possibilities of other professionals working together in the ICCO.

“*Well then it would perhaps have been nicer if I had had a bit more information about that, would have had all the possibilities. And um, look, if there were a, if it was clear to me that there is a dietician and a psychologist who work together in the context of overweight in this approach, that would be a lower threshold for me to go there than being referred to a psychologist. That would really mean to me that he belongs here in this little circle, right? Which path do you chose? He is also involved in the overweight problem. That would have helped me. If I had known...*”(Mother 2)

In the questionnaire parents were asked about referrals to other professionals by YHC or by the general practitioner last year. In Table 3 the number of referrals is shown for the families in the ICCO questionnaire group. Most children were not referred to other professionals.

#### 3.4.3. Creating a Healthy Environment for Children

Parents experienced more attention to exercising and sports during and after school time and they experienced new rules in school to promote fruit, healthy lunches and healthy birthday treats. Schools also promoted drinking water instead of taking juices from home. The interviewed parents were happy with the promotion of healthy choices by the schools. They found it important to make the environment healthier for children.

“*Well, I think the initiative is just super good and I think they should continue with it and actually expand it even more. Because at a certain point people will go along with it anyway because they have no choice, and they can’t do otherwise. And that’s just like at school, if a parent comes along and wants to treat them with crisps, well, that’s bad luck, because then they’ve just bought them for nothing.*” (Mother 5)

### 3.5. Tops and Tips by Interviewed Families

At the end of the interview, parents and children summarized what they appreciated in the support from the CP in the ICCO. They also formulated points for improvement. These points of appreciation (tops) and improvement (tips) are shown in Table 4.

Parents mostly appreciate that the CP had a good connection with the family, had enough time and focused on the parents’ needs.

According to the parents, points of improvement can be found in the follow-up and in arranging a smooth transfer to another CP if a CP gets another job. Next to that there were some specific needs from parents. One of the mothers had expected more practical support from the CP by, for example, arranging and financing more possibilities to exercise for her child with obesity. Support from a physiotherapist was only reimbursed by the health insurance for a limited number of sessions.

Another mother expressed the wish to speak with the CP without the child, because of the sensitivity of the overweight topic. She would like to get more support on how to discuss overweight with her child.

“*So, I felt supported, yes, what I find really difficult is to talk about overweight in front of [child]. Because I just don’t think that’s necessary. I think it’s good to teach her to be healthy with her weight, but I also find it very difficult to talk with her about being overweight.*”(Mother 2)

### 3.6. Comparison with Families Receiving Care as Usual

Characteristics of the children and parents who completed the questionnaire in the control group receiving care as usual (*n* = 20) were comparable with the ICCO group (*n* = 38). Parents in the control group gave a mean of 7.0 to the support of the YHC. The number of referrals to other professionals were comparable in the ICCO and control group. However, looking to other involved professionals, the dietician and pediatrician were more often involved in families in the ICCO group than in the control group.

There were no statistically significant differences in characteristics, grade, referrals and involved professionals between the ICCO and the control group.

## 4. Discussion and Conclusions

The aim of this study was to explore the experiences of parents and children with the newly implemented role of the YHC nurse as CP in the ICCO. Interviewed parents and children felt supported by the YHC nurse. They appreciated the broad perspective and the personal approach focusing on the families’ needs. Children felt understood. Contact with the CP was not very frequent. One of the points for improvement is taking care of follow-up after a period of support. Next to the YHC, families had the most contact with a pediatrician and dietician. Most children were not referred to another professional last year. Only a few parents did actually experience collaboration in the ICCO.

In the ICCO in ‘s-Hertogenbosch the YHC nurse has a role as CP. The YHC nurse works outside the hospital setting and stands beside the family, looking at what this family specifically needs. This is a completely different approach than only giving lifestyle advise or referring a family to a structured Combined Lifestyle Intervention (CLI) to solve a child’s overweight. In an ideal situation, a family receives support from a CP, and a CLI is only started if and when it fits the situation and motivation of the child and parents. Unfortunately, in most municipalities in The Netherlands, the support from a CP and a CLI are not available yet [20].

Parents felt supported and appreciated the personal approach by the CP. A ‘one size fits all’ approach is not desirable in overweight interventions, given the complexity of the lives of families. Grootens-Wiegers et al. describe that participation in a lifestyle intervention is not influenced by one specific factor but by the interplay of facilitators and barriers. Understanding the complex circumstances of participants is wanted to personalize guidance towards and execution of interventions [8].

Long-term follow-up of families is wanted. This was also mentioned in the evaluation report of Your Coach Next Door, the name of an approach quite similar to the ICCO, in Maastricht [21]. The authors of this report describe involvement of the CP for a longer period of time and mention that embedding and taking charge by the family takes time. They also describe involvement of some other professionals: 25% of the children had contact with a dietician and 13% received psychological support from a practice nurse for youth mental health [21]. In our study, slightly more children (32%) had contact with a dietician. The practice nurse for youth mental health was not mentioned as a commonly contacted professional in the ICCO in our study and such a professional is not even available in all municipalities. This shows that the network of involved professionals and collaboration between professionals can differ between municipalities according to the local situation.

Looking at the way the role of CP is fulfilled, a substantial difference between Your Coach Next Door in Maastricht and the ICCO in ‘s-Hertogenbosch can be seen. In ‘s-Hertogenbosch all 14 YHC nurses in four neighborhoods were trained to support families as a CP, whereas in Maastricht only three YHC nurses working as a CP supported 103 children with overweight and obesity in 2019–2020 [21]. Less YHC nurses spending more time per week to their role as a CP can possibly help to focus and prioritize towards family support for children with overweight and obesity, even for a longer time. In our previous study focusing on the YHC nurses in their new role, lack of time, limited capacity and having other priorities were often mentioned as barriers to completely fulfil their role as a CP [15].

The interviewed parents appreciated the role of the YHC nurse as a CP, although they mentioned also various points for improvement. Parents did not experience a lack of time by the YHC nurse, although contacts were not very frequent, and follow-up could have been organized more frequently.

CP’s and other professionals need good communication skills and have to be sensitive to the needs of the families [22]. Many parents experienced an emotional response when told about their child’s weight, ranging from a positive response to disbelief to negative feelings [23]. In our study, all interviewed parents remembered the first conversation about the overweight of their child, emphasizing the importance of a good attitude by the professional.

Also, the interviewed children appreciated the contact with the CP. They felt understood and supported. This is in line with a review evaluating the view of adolescents attending obesity interventions. It showed that support from professionals, family and peers was valued highly. Adolescents expressed prior fears of attending interventions and wanted longer term support [24]. A CP can be an easy, trusted, approachable coach for a child or adolescent with overweight or obesity, providing long-term support at low costs.

Parents did experience and appreciate increased attention for exercise and healthy food and drinks at schools. Many schools in ’s-Hertogenbosch participate in The Healthy School approach that was developed to support schools in working systematically and structurally on health and a healthy lifestyle of pupils [25]. Also, JOGG (Young People at A Healthy Weight) coordinates activities promoting a healthy lifestyle in ‘s-Hertogenbosch since 2014. JOGG is an integrated community-based approach focusing on locally embedded activities and environmental changes stimulating healthier eating patterns and more physical activity in children [26,27]. The influence of a CP highly depends on the environment in which she operates. The literature shows that the design of an environment that facilitates healthy choices is important. This applies to all types of environmental factors (physical, social, political, economic), but the involvement of parents is often mentioned. Context-related factors are consistently mentioned as beneficial for the effects of potentially effective elements of more focused interventions [28]. A CP needs other professionals and a supportive environment to work in. Hindering factors for collaboration in the ICCO, mentioned by (some) professionals are a lack of clarity about roles and responsibilities, insufficient feedback provision after referrals and lack of funding and time. Knowing each other is mentioned as an important facilitating factor for collaboration in the ICCO [15].

The implementation of the role of the CP was not completed by the time the questionnaires and interviews were done in ‘s-Hertogenbosch. Some parents only had irregular and low intensive contact with the CP. The ICCO is still developing and further implementation of the role of the CP needs attention, especially after the start of COVID-19.

### 4.1. Strengths and Limitations

Qualitative research methods are an effective way to explore the experiences of people. They are valuable in providing rich descriptions of complex phenomena. Qualitative and quantitative methods can be complementary, used in sequence or in tandem [29]. In our study both qualitative and quantitative methods were used to describe the experiences of families with the CP in the ICCO. By interviewing both the parent(s) and the child, experiences from both perspectives were collected. The stories from the interviews of the families helped us to interpret the data from the questionnaires.

However, our results may be biased, since interviewed parents seemed to appreciate the YHC more than the parents who did not participate in an interview. Only a small group of families was interviewed and apparently not a proportional reflection of the whole ICCO group, which is a limitation for the generalizability of our results. None of the parents with an appreciation grade below 7 responded to an invitation for an interview. As a result, their perspective and points of criticism may have been missed.

More families with a migration background were interviewed in comparison with the ICCO questionnaire group. Relatively more interviewed children were obese and interviewed parents had more contact with the YHC nurse and other professionals. The interviewed group could possibly have had more benefit from the support of a CP than a group with less involvement of other professionals.

It would have been interesting to interview families in the control municipalities too about their experiences with the support of a YHC nurse receiving care as usual, without training as a CP and without an ICCO implemented in the municipality. The restrictive rules because of lockdown made more face-to-face interviews after March 2020 impossible. The last interview in the ICCO group was done by video phone call, which appeared not optimal and did not provide much new information.

### 4.2. Conclusions

Parents and children appreciated the new role of the YHC nurse as CP in the ICCO. Strengthening the promising role of the CP in the ICCO is recommended. The majority of parents felt supported and understood by the CP and appreciated the broad perspective taken and the adaptation of the support to the specific families’ needs. Implementation of this new approach, however, also needs more attention. Special attention is needed for professional follow-up and continuity which is to be communicated properly with parents. In addition, the collaboration between professionals in the care network for children with obesity should be further intensified.

### 4.3. Recommendations

Based on the findings of our study regarding the new role of the Youth Health Care nurse as Coordinating Professional in the ICCO approach for prevention of childhood overweight and obesity, some recommendations for further development of this approach can be formulated.

Use the feedback of the experiences of parents and children for further development of the CP’s role in the ICCO.Retain and further invest into the supportive attitude of the CP, the approachability, the broad perspective and the adaptation to the needs of the family which appear to be the most innovative and appealing to families with children with overweight and obesity.Follow-up by the CP should be better organized and agreed upon with parents.Make the role of CP more explicit and position it more clearly, also towards parents. Explain what the CP can do for families in addition to the ‘normal’ tasks of YHC.Offer parents the possibility to speak to the CP alone, without the child present.Explain to parents what the ICCO is, and which professionals work together. What kind of low threshold possibilities does this offer?Look carefully in which target group the support of a CP is most appropriate. The CP can focus on families with complex problems, where multiple professionals are involved.Discuss whether it is wanted that all YHC nurses work as a CP or that some YHC nurses specialize to fulfil this role.

## Figures and Tables

**Table 1 ijerph-19-05797-t001:** Characteristics of participants; interviews and questionnaire.

	Interview Group (*n* = 8)	Questionnaire ICCO Group (*n* = 38)
Child age, year, mean (range)	10.8 (8–12)	9.3 (5–13)
Child sex, boys, *n* (%)	3 (37.5)	17 (44.7)
Obesity ^(1)^, *n* (%)	6 (75.0)	15 (39.5)
Both parents not born in the Netherlands, *n* (%)	5 (62.5) *	6 (15.8)
Mother with lower vocational education or less, *n* (%)	3 (37.5)	7 (18.4)
Mother with middle vocational education or more, *n* (%)	3 (37.5)	11 (28.9)
Parents have (some) problems to make ends meet (yes/no), *n* (%)	4 (50.0)	12 (31.6)
Child lives with both parents (yes/no), *n* (%)	6 (75.0)	29 (76.3)
Number of siblings, mean (range)	1.3 (0–3)	1 (0–2)

* Statistic significant difference, *p* < 0.05; ^(1)^ BMI measured by YHC professionals according to IOTF standards [19].

**Table 2 ijerph-19-05797-t002:** Families’ contact with other professionals last year.

Professional	Interview Group (*n* = 8)		Questionnaire ICCO Group (*n* = 38) *	
Pediatrician	3	37.5%	8	21.1%
Social Worker	2	25.0%	3	7.9%
Social Neighborhood Team	1	12.5%	1	2.6%
Dietician	1	12.5%	12	31.6%
Physiotherapist	2	25.0%	4	10.5%
Combined Lifestyle Intervention	0	0.0%	1	2.6%
Lifestyle coach	1	12.5%	1	2.6%

* Including six out of eight families who were interviewed.

**Table 3 ijerph-19-05797-t003:** Number of children referred to other professionals in the ICCO questionnaire group (*n* = 38) by Youth health Care and General Practitioner.

	Number of Referrals by Youth Health Care in 38 Children	Number of Referrals by General Practitioner in 38 Children
Pediatrician	4 (10.5%)	5 (13.2%)
Social Worker	1 (2.6%)	0 (0%)
Social Neighborhood Team	1 (2.6%)	0 (0%)
Dietician	6 (15.8%)	1 (2.6%)
Physiotherapist	1 (2.6%)	1 (2.6%)
Combined Lifestyle Intervention	1 (2.6%)	1 (2.6%)
No referral	23 (60.5%)	30 (78.9%)
Other referral	3 (7.9%)	2 (5.3%)

**Table 4 ijerph-19-05797-t004:** Points of appreciation (tops) and improvement (tips) by parents and children.

Interview Participants *	Tops	Tips
1. Mother, father, girl (11 years old)	− Not authoritative − Enough time − He doesn’t make you come back every three months, which is nice − They do their best − It is also up to us − Always helped, nicely done	
2. Mother	− No judgement − Enough time − Looking beyond eating/drinking/exercising − Focusing on the question of the parent	− More information on the development of obesity − More information about network professionals − Time with parents without child − Information about how to talk about overweight with your child
3. Mother, father, girl (12 years old)	− Everything − Makes a game to weigh and measure − They listen and we listen to them	− It may be too difficult for some parents − Never seen/heard of an interpreter − Be aware of possible language barrier in some families − Language is important
4. Mother, girl (11 years old)	− Understanding − Making arrangements	−
5. Mother, boy (11 years old)	− Understanding each other − Connection − Knowing our background, knowing your family − I like to do things by myself, sometimes taking over is nice (by social work)	− When the YHC nurse stopped, finishing together would have been nice − Do a transfer to another CP
6. Mother, girl (10 years old)	− Good to come to know how the past year has gone, overweight has diminished	− Invite regularly for appointments − Promote drinking water in the classroom
7. Mother, boy (10 years old)	− Lots of information discussed from pregnancy until now − Listening well − Contact with other involved professionals − Does her best	− Make a brochure for children − Make a video for parents and children about what if you don’t eat healthy − More practical support is wanted

* Tops and tips were collected in 7 out of 8 interviews.

## Data Availability

The data presented in this study are available on request from the corresponding author. The data are not publicly available due to privacy reasons.

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
