# Peer review of "Parents’ and Children’s Experiences with a Coordinating Professional in Integrated Care for Childhood Overweight and Obesity—A Novel Dutch Approach"

_ijerph, 2022, doi:10.3390/ijerph19105797_

Round 1

Reviewer 1 Report

Congratulations to the authors for their interesting work. In my view it can be published in the present form.

Author Response

Reviewer 1: congratulations and article can be published in the present form.

We thank reviewer 1 for the positive feedback.

Reviewer 2 Report

Title: Parents’ and children’s experiences with a Coordinating Professional in Integrated Care for Childhood Overweight and Obesity. A novel Dutch approach.

The study aims to explore the experiences of parents and children with the role of Youth Health Care nurse as CP in the integrated care for childhood and obesity. The article covers an important area of health in pediatric population. However, the presented results are somewhat disappointing. The study population is very small and authors present the individual opinions of parents and children with the PC. The data were collected according to different manners and using open question questionnaire and presented as discussion with responders.

            In my opinion the finding that: Most of the families did not have very intensive or frequent contact with their CP last year (lines 166-167) indicates a weakness in the CP's work.

This is also supported by the finding that for some families the last contact between parents and the CP was a long time ago without any planned follow-up (lines 174 -175).

I also have doubts about the evaluation of YHC nurse. What do the authors assess by asking the following question: “If you had to give the YHC nurse a grade, what grade would you give him? (line 189)

And finally, the majority of results, including those related to children’s experiences with the CP are presented using not numbers but statements: more interviewed children…, some children felt motivated.    

In conclusion, the article is methodologically inadequate, does not present significant accomplishments, and should not be published in this format

Author Response

We thank the reviewer for the constructive feedback.

Reviewer 3 Report

Dear Authors

Thank you for the opportunity to review this article. Below are some comments for improving the presentation quality:

In the abstract section:

  • The method is not clear. I suggest including the data analysis strategies;
  • I would suggest presenting participants' data in the results section;
  • I recommend adding a section entitled 'findings' to highlight the study data.

Keywords section:

I suggest adding at least two keywords highlighting the method of conducting the study (qualitative interviews; online questionnaire). This makes it easier to cite your study.

2. "Materials & methods

Overall, the description of the method adopted needs to be improved. I suggest following the COREQ (COnsolidated criteria for REporting Qualitative research) Checklist (https://www.equator-network.org/reporting-guidelines/coreq/) for reporting the study clearly.

In addition, it lacks an accurate description of the data analysis strategy. I would suggest including this section for both methods adopted in the study.

I suggest describing strategies you adopted for ensuring method rigor and validity, as well as the credibility of reported findings.

Section 2.1 "Design":

Unclear. I would suggest structuring the description of the study design better. In addition, the choice of study design needs justification. It is not clear whether it is a mixed-methods study. At the same time, it seems to be a trial with an intervention group and a control group: however, this description is confused and not clear.

Section 2.2: 'Data collection':

What you reported in the first sentence of this section should be placed within a section on ethical issues. In addition, ethical aspects (e.g. informed consent) need accurate description, as children with health problems were involved.

What you reported in the second sentence, within recruitment strategies.

What is required in the data collection is a description of "the data collection strategies" you adopted, justifying the choices, and explaining well the construction or adoption of the instruments used (interviews, questionnaire).

Section 2.2.1 "Interviews".

As for the abstract, I suggest that you report your recruitment results data in the "results" section (the same is true for the duration of the interviews.). In addition, it would be helpful if you provided the interview guide, as well as a brief description of the themes explored during the interviews: to allow readers to better understand your study.

Section 2.2.2 "Questionnaires".

As with the abstract, I suggest reporting the results data in the "results" section.

3. Results

Table 1 shows data in a way that does not appear inclusive. Here you should report the participants' characteristics, not your interpretation of their level of education (for example). I suggest reporting this type of data in a more descriptive, less interpretive way. You have the opportunity to state your considerations regarding socio-demographic issues in the discussion section.

I would suggest adding a section entitled "findings" for describing themes that emerged during the data analysis. In this section, it is necessary to explain how many (and which) themes emerged. The same applies to the abstract section.

I suggest a more detailed description of the impact of this intervention on children. Which descriptive features were identified from the interviews?

4.2 "Conclusions"

This section is poorly reported. I suggest expanding it.

4.3 "Recommendations

I suggest starting this section with an introductory sentence.

Author Response

(The authors gave the same response as above.)

Round 2

Reviewer 2 Report

I appreciate the authors' effort to improve the article, but I think it has low scientific value. 

Reviewer 3 Report

Dear authors,

I appreciate your acceptance of my suggestions. The article has improved from a methodological point of view.

Kind regards